# Exploring Large MAF Transcription Factors: Functions, Pathology, and Mouse Models with Point Mutations

**DOI:** 10.3390/genes14101883

**Published:** 2023-09-27

**Authors:** Mitsunori Fujino, Masami Ojima, Satoru Takahashi

**Affiliations:** 1Department of Anatomy and Embryology, Faculty of Medicine, University of Tsukuba, Tsukuba 305-8575, Ibaraki, Japan; mitsunori1217fujino@icloud.com (M.F.); mk03j341@md.tsukuba.ac.jp (M.O.); 2Ph.D. Program in Human Biology, School of Integrative and Global Majors, University of Tsukuba, Tsukuba 305-8575, Ibaraki, Japan; 3Laboratory Animal Resource Center, University of Tsukuba, Tsukuba 305-8575, Ibaraki, Japan; 4Life Science Center, Tsukuba Advanced Research Alliance (TARA), University of Tsukuba, Tsukuba 305-8575, Ibaraki, Japan; 5International Institute for Integrative Sleep Medicine (WPI-IIIS), University of Tsukuba, Tsukuba 305-8575, Ibaraki, Japan; 6Transborder Medical Research Center, Faculty of Medicine, University of Tsukuba, Tsukuba 305-8575, Ibaraki, Japan

**Keywords:** large MAF transcription factors, c-MAF, MAFA, MAFB, point-mutation model mouse

## Abstract

Large musculoaponeurotic fibrosarcoma (MAF) transcription factors contain acidic, basic, and leucine zipper regions. Four types of MAF have been elucidated in mice and humans, namely c-MAF, MAFA, MAFB, and NRL. This review aimed to elaborate on the functions of MAF transcription factors that have been studied in vivo so far, as well as describe the pathology of human patients and corresponding mouse models with c-MAF, MAFA, and MAFB point mutations. To identify the functions of MAF transcription factors in vivo, we generated genetically modified mice lacking c-MAF, MAFA, and MAFB and analyzed their phenotypes. Further, in recent years, c-MAF, MAFA, and MAFB have been identified as causative genes underpinning many rare diseases. Careful observation of human patients and animal models is important to examine the pathophysiological mechanisms underlying these conditions for targeted therapies. Murine models exhibit phenotypes similar to those of human patients with c-MAF, MAFA, and MAFB mutations. Therefore, generating these animal models emphasizes their usefulness for research uncovering the pathophysiology of point mutations in MAF transcription factors and the development of etiology-based therapies.

## 1. Introduction

Medical technology has rapidly advanced in recent years, leading to the identification of the causative genes for many conditions, including cardiovascular disease, diabetes, and malignancies, as well as more than 7000 rare diseases. Despite the limited number of patients with rare diseases and challenges in identifying effective treatment strategies, rapid advances in genome editing tools have accelerated the knowledge of human disease studies owing to our ability to generate diverse genetically modified animal models. Given the advances in gene editing tools, these animal models are gradually progressing toward mimicking human diseases by introducing human pathogenic mutations into their genomes, along with conventional gene knockouts.

The roles of large musculoaponeurotic fibrosarcoma (MAF) transcription factors, including c-MAF, MAFA, and MAFB, in various organs and tissues have been elucidated in both knockout and conditional knockout mice. In addition, point mutations in these genes have recently been reported to cause various rare diseases, such as congenital cataracts, Aymé–Gripp syndrome, multicentric carpotarsal osteolysis (MCTO), Duane syndrome, and focal segmental glomerulosclerosis (FSGS). Therefore, considering that large MAF transcription factors are highly conserved between humans and mice, mouse models that incorporate human pathogenic mutations into these genomes are crucial to understanding the pathophysiological mechanisms and therapeutic strategies for rare diseases.

This review includes information about existing mouse models, such as those showing the phenotypic features consequent of point mutations in large MAF transcription factors except for NRL because, to date, no reports of genetically modified mice having the same point mutation in NRL have been documented. These model mice show phenotypes similar to those in humans with well-known diseases such as diabetes and nephropathy and rare conditions such as cataracts. Despite some disparities, mouse models with point mutations in large MAF transcription factors, including c-MAF (p.Thr58Ile), MAFB (p.Leu239Pro, p.Pro59Leu), and MAFA (p.Ser64Phe) mutations, simulate several features of human diseases with relatively high fidelity. This makes them useful for studies aimed at gaining a better understanding of cellular pathophysiological mechanisms and developing etiology-based therapies.

## 2. Large MAF Transcription Factors

MAF transcription factors are detected as cellular homologs of v-MAF, an oncogene that was discovered in the avian transforming retrovirus, AS42. The virus causes musculoaponeurotic fibrosarcoma in chickens [1]. The transcription factors, which are called large Maf proteins, consist of a family of transcription factors characterized by a transcriptional activation domain, a basic region required for DNA binding, and a typical bZip structure (bZip domain) which is a motif for protein dimerization and DNA binding (Figure 1A). There are four types of large MAF transcription factors, namely c-MAF, MAFA, MAFB, and NRL; in addition, they have been reported in mice and humans [2,3]. These four transcription factors are thought to be involved in the formation of homodimers and binding to DNA sequences called Maf recognition elements (MAREs) to stimulate the transcriptional activity of nearby target genes. Although the MARE sequence is 13 or 14 bp, it has been reported that half of the sequence (half MARE) activates target genes (Figure 1B) [4]. MAF transcription factors are involved in the important functions of several distinct developmental processes, establishment of the function of cells, tissues, and organs, cell differentiation, and maintaining the function of cells and organs. In recent years, various diseases have been reported in human patients with point mutations in c-MAF, MAFA, and MAFB. In the review, we organize the up-to-date published data regarding the role of these transcription factors, c-MAF, MAFA, and MAFB, using model animals and the three MAF transcription factors in point-mutation model mice.

## 3. Roles of c-MAF

c-MAF is the cellular homolog of v-MAF. It plays an essential role in the IL-4 expression in helper T cells (Th) of the immune system [7]. Several previous studies have demonstrated the significance of c-MAF in the regulation of the functions of various immune cells, particularly T cells. For example, the scaffold protein CARMA1 and IKKβ, two essential regulators of the transcription factor nuclear factor κB (NF-κB), activate c-MAF expression. Through the stimulation of the T-cell receptor, increased c-MAF expression results in the production of cytokines [8]. The loss of c-MAF induces a defect in the number of Th-17 and T cells, suggesting that c-MAF controls the expansion of both Th-17 and T cells [9]. Subsequently, transcriptome analysis has shown that c-MAF plays an essential role in the differentiation of Th17 [10]. In the case of macrophages, c-MAF controls the expression of IL-10, which is involved in the differentiation of regulatory T cells [11]. In addition to immune cells, abnormal structures of the eye lens have also been identified in *c-Maf* homozygous-deficient mice. For example, a detailed analysis of the lens in *c-Maf* homozygous knockout mice confirmed that there was no expression of CRYSTALLIN, which is a structural protein found in the lens, due to the presence of the MARE sequence in *Crystallin* genes. These results suggest that c-MAF plays an important role in lens formation by directly regulating the gene expression of *Crystallin* [12,13]. Furthermore, c-Maf is regulated by p53 together with Prox-1, and as a result, the expression of various *Crystallin* genes is regulated [14]. In the case of cell differentiation, c-MAF plays an essential role in the differentiation of lens fiber cells to lens epithelial cells, resulting in epithelial cells spreading in the anterior and posterior lens [15]. Other MAFs, such as MAFA and MAFB, are not required for lens fiber cell differentiation [16]. Furthermore, fetal analysis identified the important role of c-MAF in bone formation [17].

The roles of c-MAF in the immune system and developmental stages have mainly been discovered through studies on *c-Maf* homo-deficient mice. These mice are embryonic lethal due to an abnormal formation of blood islands in the liver in the embryo [18]. To address this issue, conditional knockout mice using the CRISPR-Cas9 genome editing tool were generated, and their functions in the adult stage were clarified. When adult c-MAF conditional knockout mice were systemically deficient in c-MAF, they survived; however, they developed cataracts owing to lens maintenance failure. Detailed lens analysis revealed that the abnormal lens structure and onset of cataracts were caused by the inability to differentiate from lens epithelial cells into lens fiber cells. To our knowledge, this is the first study to demonstrate the role of c-MAF in adults [19]. Subsequently, other studies have demonstrated inhibition of the development of the liver sinusoid by specific deletion of c-MAF in endothelial cells, which induces aberrant expansion of postnatal liver hematopoiesis, accelerates excessive proliferation of sinusoidal cells during the postnatal period, and worsens sensitivity of pro-fibrotic effects in the liver to chemical insults [20]. Moreover, the mechanistic roles of c-MAF in the recovery from acute injury in the intestine promote the differentiation of epithelial cells and the importance of crosstalk between differentiated tuft cells and enterocytes to accelerate long-term intestinal accommodation [21]. While these two studies suggested a requirement for c-MAF in organ injury, we report a protective effect of c-MAF deficiency against diabetes and diabetes-induced nephropathy through c-MAF regulation of the glucose transporters SGLT2 and GLUT2, which are expressed in the proximal tubules of kidneys [22]. In summary, c-MAF deficiency is interesting and has the potential to decrease susceptibility and provide protection against injury and diseases.

## 4. c-MAF Point Mutation in Human Patients

Conditions caused by a point mutation in c-MAF were first discovered in 2002, wherein patients showed congenital cataracts and iris hypoplasia [23]. In 2006, patients with pulverulent cataracts, cataracts with microcornea, iris coloboma, and congenital cataracts were reported [24,25]. Moreover, the vibratory sensation was reduced in patients with the same arginine-to-proline mutation in the DNA-binding region of c-MAF as in human cases discovered in 2002 [26]. All these cases were caused by the inability of c-MAF to bind to the MARE sequences of the target genes due to mutations in the DNA-binding site of c-MAF, in which arginine was changed into proline (Figure 2 and Appendix A).

For mutations in the transcriptional activation domain, c-MAF was detected as the causative gene for Aymé–Gripp syndrome resulting in congenital cataracts, deafness, mental retardation, epilepsy, and skeletal dysplasia [27]. Since then, various conditions such as dental abnormalities, calcifications of the bilateral putamen nucleus, pericardial effusion, femoral neck fracture, and exostosis of the distal phalanx have been reported [28,29,30,31]. Here, a case of Aymé–Gripp syndrome was reported to have a mutation in the phosphorylation residues of the transcriptional activation region, such as Ser54, Thr58, Pro59, Ser62, and Pro69 (Figure 3 and Appendix A). Interestingly, this abnormal phosphorylation has been reported in cases of human disease mutations of MAFA and MAFB, both in heterozygous and homozygous individuals. Although only one MAFA mutation has been identified, several MAFB mutations have been identified in the transcriptional activation domain, including p.Ser54Trp, p.Pro59Leu, p.Thr62Ile/Pro, p.Pro63Leu/Arg, p.Ser65Ile, p.Ser66Cys, p.Ser69Leu, p.Ser70Leu/Ala, and p.Pro71Ser (Appendix A). These phosphorylation residues are preserved in human MAF transcription factors, c-MAF, MAFA, and MAFB, and are strongly associated with the development of human diseases.

## 5. Model Mouse of c-MAF Human Point Mutation

To the best of our knowledge, we successfully generated a c-MAF point-mutation mouse for the first time [32]. To investigate the underlying mechanisms of disease onset and identify previously unknown diseases in human patients, a mouse model with a mutation in c-MAF on the GSK3 phosphorylation site, i.e., p.Thr58Ile was generated using the CRISPR-Cas9 system.

The results of sequencing, PCR analysis, IHC staining, and the increased levels and expansion of c-MAF in *c-Maf* heterozygous mice were compared to those for *c-Maf* wild-type mice. Moreover, to quantify both RNA and protein expression levels of *c-Maf* in the kidneys, quantitative PCR (qPCR) analysis and Western blotting were performed. These results showed no significant difference between *c-Maf* heterozygous and wild-type mice in RNA levels, but we observed denser unphosphorylated bands in the Western blotting analysis between *c-Maf* heterozygous and wild-type mice. These results suggest that the increased unphosphorylated protein expression of c-MAF in *c-Maf* heterozygous mice, compared to that in wild-type mice, may be attributed to impaired phosphorylation of c-MAF and declined degradation of the c-MAF proteins with the mutation. Therefore, these results confirm the successful insertion of the *c-Maf* mutated allele into the *c-Maf* target region, resulting in elevated expression of c-MAF in *c-Maf* heterozygous mice, compared to that in wild-type mice.

Our experimental model mouse exhibited phenotypes similar to those observed in human cases, including lens abnormalities, growth retardation, abnormal skull morphology, and short stature. It should be noted that a phenotype was found that was not observed previously in human patients. First, the *c-Maf* homozygous mutation mouse embryos became embryonic lethal, similar to the *c-Maf* homozygous knockout embryos. The *c-Maf* homozygous mutations induce changes in appearance such as shorter limb sizes and growth retardation of the body in the embryonic stage. This underpins the reason why homozygous patients have not yet been identified. Another difference was in the brain weight between c-MAF-mutant mice and control mice. Although patients with c-MAF mutations exhibit intellectual and developmental disabilities, the reason for this has not yet been elucidated. Therefore, this mouse model may be useful for future investigations to assess whether this difference is due to a smaller skull size, a decrease in brain weight, or other underlying factors that are currently unclear. Although Aymé–Gripp syndrome is rare, our experimental murine model may be used to discover previously unreported syndromes in patients and serve as an ideal model for establishing new treatments. Therefore, these kinds of point mutation mouse models are valuable because they can provide novel functional insights into both humans and mice and can be used to analyze the functional roles of target genes.

## 6. Roles of MAFA

MAFA is detected as a transcription factor inducing the development of the lens of eyes in the epidermis of chickens [33]. Further studies showed that MAFA activates the insulin gene C1 element through the binding of the promoter site, contributing to the increased expression, function, and differentiation of insulin in β cells of the pancreas [34,35]. MAFA coordinates with MAFB, a transactivator of cells, acting on the glucagon gene G1 element, and in conjunction with other transcription factors and related genes to induce the generation and differentiation of β cells [36,37]. In subsequent studies, MAFB was identified in both α and β cells during the early developmental stage. Following a reduction in MAFB expression, MAFA is mainly expressed instead of MAFB [38,39,40]. To identify the function of MAFA in mice, we generated *Mafa*-knockout ICR mice. However, while *Mafa* homo-deficient mice were born normally, in contrast to c-MAF-knockout mice, they subsequently developed mild fasting hyperglycemia. Further, the mice exhibited abnormal glucose, and histological analysis of the islets in the pancreas revealed abnormal strictures in the adult stage. Gene expression analysis identified various genes, such as *Ins1*, *Ins2*, *Pdx1*, *Glut2*, and *Glanuphilin*, as target genes of MAFA [41,42]. Although the expression profile of MAFA indicates that it is highly expressed in the cerebellum, skeletal muscle, thyroid, and testes, the detailed roles of MAFA remain unclear, except in the lens and pancreatic islets.

## 7. MAFA Point Mutation in Human Patients

In 2018, two unrelated families with point mutations in the transcriptional activation domain of the *MAFA* gene were reported to have familial diabetes and insulinoma. In addition, the report showed that insulinomatosis was more frequent in females and diabetes was more frequent in males. A missense mutation of MAFA (p.Ser64Phe, c.191C > T) induces the phenotypes of both insulinomatosis and diabetes; in addition, the mutation was found to decrease phosphorylation within the transactivation region of MAFA, similar to a mutation of c-MAF on a GSK3 phosphorylation site (Figure 3 and Appendix A). Further, the protein stability of MAFA profoundly increased under both high and low concentrations of glucose in β-cell lines. Therefore, phenotypes in human cases with the missense mutation of MAFA reflect both the oncogenic capacity of MAFA and its important role in the activity of pancreatic β-cells [43].

## 8. Model Mouse of MafA Human Point Mutation

In 2021, since the number of human patients with MAFA mutation is small and mechanistic studies in humans are difficult, CRISPR-Cas9 gene editing was used to generate an experimental mouse model harboring the same pathogenic change of a single base pair as human patients [44]. This heterozygous mutant male showed impaired glucose tolerance, whereas heterozygous mutant female mice showed slight hypoglycemia. Surprisingly, protein expression of MAFA in four-week-old heterozygous mutant male mice was significantly increased one week before the onset of glucose intolerance. In contrast, heterozygous mutant female mice did not show elevated expression. This finding may explain the different phenotypes observed in heterozygous mutant male and female mice.

Notably, differences exist between c-MAF- and MAFB-mutant mice and MAFA-mutant mice in terms of their proclivity to develop insulinoma, a typical disease in humans. Although female mutant mice displayed slight hypoglycemia and the amelioration of glucose tolerance, the previously mentioned study did not confirm whether insulinoma development occurred or the reasons for it. Further, c-MAF- and MAFB-mutant mice exhibited similar phenotypes to human patients, suggesting the need to create conditions conducive to insulinoma development. Nevertheless, these mice are useful experimental models for diabetes research due to their abnormal β cells.

## 9. Roles of MAFB

The *Mafb* gene was detected in Kreisler (Kr) mice, showing turning behavior resulting from impairment of the development of the inner ear by radiation. Further study indicated that MafB plays an important role in segment formation in the hindbrain; in addition, the absence of MafB induces abnormal formation of the inner ear [45]. MAFB was isolated through homology screening for v-MAFs [46] and was found to inhibit the function of ETS1 and the differentiation of erythroid cells in the hematopoietic system in chickens [47]. Moreover, overexpression of MAFB stimulates the differentiation of both macrophages and monocytes [48]. Notably, a previous study showed that *krml* homozygous mutant mice develop nephrotic syndrome resulting from glomerular podocyte impairment. In addition, the study highlighted that MAFB was shown to be essential for the differentiation of the epithelial cells of the podocytes during the final stages of development [49].

To elucidate the roles of MAFB, we established *Mafb*-deficient mice through targeted gene deletion. However, the *Mafb* homozygous knockout mice died immediately after birth since the newborn mice suffered from respiratory failure caused by hypoplasia of the neuronal respiratory center [50]. Subsequently, we generated *Mafb-Gfp* knock-in/knockout mice by replacing the *Mafb* gene with the *Gfp* gene. Similar to the previous study, *Mafb-Gfp* knock-in/knock-out mice died immediately after birth due to hypoplasia of the hindbrain, and kidney and respiratory failure. However, the number of macrophages expected to increase, as in the previous study [48], was normal. Mafb knockout suppressed F4/80 expression in mature macrophages, resulting in renal dysgenesis with abnormal podocyte differentiation and tubular apoptosis [51]. Subsequently, it was reported that although there was no abnormality in the differentiation of macrophages, there was an abnormality in the formation of actin in macrophages following the deletion of MAFB [52]. Furthermore, *Mafb* heterozygous mice showed abnormal parathyroid gland formation, while homozygous-deficient mice did not produce parathyroid hormone [53]. These results suggest diverse roles of MAFB in mice. In addition, it was reported that MAFB is involved in various biological processes, including the myeloid commitment of hematopoietic stem cells, growth of macrophages [54,55], generation of the thymus [56], creation of hair cuticles [57], differentiation of skin keratinocytes [58], formation of the urethra [59], and lymphangiogenesis [60].

Although the various roles of MAFB in the embryonic stage have been elucidated, due to the immediate post-birth lethality of *Mafb* homozygous knockout mice, we generated conditional MAFB-knockout mice to analyze their functions in mice. We found that the expression of apoptosis inhibitor of macrophages (AIM) was controlled by MAFB. This result suggests that MAFB plays an important in atherosclerotic lesion development by inhibiting foam cell apoptosis [61]. Furthermore, MAFB regulates the phagocytic function of macrophages by regulating the expression of C1q, which is a complement component essential for the phagocytosis of foreign substances and dead cells, and the expression of MSR1, which is a scavenger receptor [62,63]. MAFB is necessary for the creation and maintenance of podocytes in kidneys at the embryonic [49] and adult stages, respectively [64]. In 2007, it was reported that MAFB is expressed in α and β cells of the mouse fetal pancreas and *Mafb* homozygous-deficient mice show suppressed development of α and β cells in the pancreas during the embryonic stage [39]. Although MAFA is essential for the expression of insulin in β cells of the pancreas, using conditional knockout mice, it was revealed that MAFB also contributes to pancreatic β-cell development during the embryonic phase. While MAFB is inactive under normal conditions, it functions under pathological conditions, underpinning its importance in the development and maintenance of cell functions [65,66].

## 10. MAFB Point Mutation in Human Patients

Since c-MAF plays important roles in lens, bone, liver, and kidney formation, and patients with c-MAF mutations show many disorders, such as congenital cataracts, deafness, skeletal dysplasia, dental abnormalities, epilepsy, mental retardation, calcifications of the bilateral putamen nucleus, pericardial effusion, exostosis of the distal phalanx, and femoral neck fracture [21,22,23,24,25], it was considered that MAFB mutations also cause various diseases. Since *Mafb* homozygous knockout mice die immediately after birth due to respiratory failure, it was thought that *Mafb* heterozygous mutant patients might have abnormalities in their respiratory organs, followed by several other organs, such as the ears, parathyroid glands, and kidneys, as demonstrated in MAFB-knockout and conditional knockout mice.

In 2012, the results of whole exome sequencing of human cases identified *MAFB* as the causative gene for multicentric carpotarsal osteolysis (MCTO), which is a syndrome showing the dissolution of the bones of the palm and foot [67]. Similar to c-MAF and MAFA mutation cases, MCTO human patients harbor mutations at the phosphorylation site of the transcriptional activation region, resulting in c.176C > T (p.Pro59Leu), and abnormal phosphorylation inhibiting the degradation of MAFB (Figure 3 and Appendix A). In addition to abnormal bone formation, MCTO cases show hereditary FSGS.

Following this discovery, not only patients with mutations in the activation domain but also those with mutations in the binding region were found to be similar to c-MAF. One previous study in 2016 detected MAFB as the causative gene of Duane syndrome, which is characterized by abnormal eye abduction. In Duane syndrome, the cleaved MAFB protein may be encoded by the mutated *MAFB* gene [68]. Clinically, we identified a Japanese patient with a novel MAFB mutation harboring a rare heterozygous substitution (p.Leu239Pro), presenting Duane syndrome and FSGS (Figure 2 and Appendix A). The mutation is a highly preserved leucine-to-proline substitution in the DNA-binding region of MAFB, rendering it incapable of binding to the MARE sequence. Finally, podocytes from neonatal mice with the p.Leu239Pro mutation exhibited impaired differentiation. Therefore, mutations of MAFB impair the generation and/or inhibit the sustenance of abducens neurons, the inner ear, and podocytes in the kidneys [69].

## 11. Model Mouse of MAFB Human Point Mutation

We introduced mutations similar to those found in human patients (p.Pro59Leu and p.Leu239Pro) into mice using genome editing. We generated two types of MAFB mutation mouse models and demonstrated that pathological changes similar to those found in human patients were induced.

A mouse model with MCTO was generated using CRISPR-Cas9 gene editing [70]. *Mafb* homozygous mutant mice exhibited MCTO and nephropathies such as glomerular sclerosis and renal failure, developmental defects in body weight, and high levels of urine albumin and creatinine, which are similar to the symptoms found in MCTO human cases. In the clinical setting, variability among patients in diagnosis has been reported; however, this has occurred for unknown reasons [71]. Thus, it is necessary to establish other MCTO mouse models with various mutations to evaluate target genes surrounding MAFB and clarify the implications of these reasons. In MCTO patients, glomerular sclerosis and renal failure frequently occur, and MCTO is diagnosed in the later stages of renal failure. Investigating its indications at earlier stages in *Mafb* homozygous mutant mice with MCTO can aid in developing diagnostic biomarkers to prevent the progression of nephropathy in patients with MCTO. To identify the pathogenicity of the heterozygous substitution of p.Leu239Pro, an experimental model mouse carrying the MAFB p.Leu239Pro mutation was established using the CRISPR-Cas9 system [69]. *Mafb* homozygous mutant mice died within 48 h after birth, and *Mafb* homozygous newborn mice showed defective differentiation of podocytes resulting from the impairment of foot processes. In contrast, *Mafb* heterozygous mutant mice could stay healthy without any abnormal podocyte formation or kidney function. In further analyses using MAFB-mutant mice in 2020, we found that the MAFB p.Leu239Pro mutation was essential for developing various cells and organs, such as macrophages, α and β cells in the pancreas, podocytes in the kidneys, the parathyroid, and the inner ear [72], consistent with the findings of several other studies using knockout mice and conditional knockout mice.

Current methods used to treat MCTO nephropathy are invasive and cause severe side effects, with limited application. The usefulness of the two mouse models enabled us to develop alternative treatments, especially for MCTO nephropathy. Moreover, MCTO pathogenesis can be elucidated using an MAFB-mutant mouse model, which may contribute to the generation of novel treatments for MCTO and nephropathy.

## 12. Discussion and Conclusions

In this review, we provide an update concerning the roles of MAF transcription factors in the generation and maintenance of various organs; in addition, we report them as causative genes for point-mutation diseases in humans such as diabetes, insulinoma, MCTO, nephropathies such as glomerular sclerosis and renal failure, Duane syndrome, FSGS, pulverulent cataracts, cataracts with microcornea, iris coloboma, congenital cataracts, and Aymé–Gripp syndrome. In addition, we discuss phenotypes similar to those of human patients exhibited by animal models of the MAF transcription factors, c-MAF, MAFA, and MAFB. In basic research using experimental animals for clinical studies, close observation of patients and animal models is important for studying the pathophysiological mechanisms of rare diseases and developing therapeutic strategies. In other words, short-term observation using experimental animal models is necessary and valuable to elucidate the unknown mechanisms of lesions in human cases in comparison to long-term clinical observation. For example, in c-MAF mutation cases, a previous study in 2015 reported that a 43-year-old c-MAF point-mutation human case showed a high concentration of albumin in urine resulting in mesangiocapillary glomerulopathy [27]. The report adds clinical value to pathogenic variants of c-MAF with respect to changes in renal function. The use of experimental animals makes it possible to identify various phenotypes and greatly reduces the long-term observations required in clinical practice. Therefore, experimental model mice can help detect the details of each symptom that has rarely been observed in human cases with the mutation of the three MAFs and identify the clinical spectrum of MAF transcription factors in the future.

Moreover, experimental mouse models are ideal organisms for clinical research. First, since biopsies are carried out only for purposes of diagnosis and therapy invasively, it is not easy to detect specific pathological changes in human cases. Furthermore, it is almost impossible to identify symptoms that happen during the human embryonic stage. Therefore, our MAF-mutant mouse models are valuable for detecting specific lesions that are unknown, as well as the timing of the function of large MAFs in each cell and organ where the lesions occur.

The disease onset process and molecular mechanisms in human patients with MAF transcription factors and point-mutation model mice remain unclear as model mice have only recently been produced. MAF transcription factors are expressed across multiple organs; thus, it is expected that new diseases will be identified in the future in addition to the novel information presented in this review. Furthermore, in relation to carcinogenesis, it has been shown that MAF transcription factors, including c-MAF, MAFA, and MAFB, are importantly implicated in the development of multiple myeloma and insulinoma, and they are hypothesized to also be essential for the generation of other cancers. We are looking forward to further analyses of human cases. We intend to assess the phenotypic variations and mechanism of treatment in greater detail in the future.

## Figures and Tables

**Figure 1 genes-14-01883-f001:**
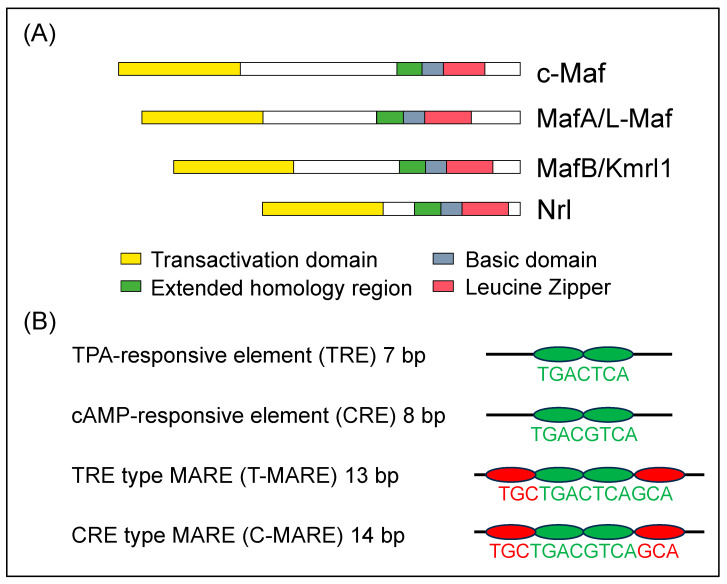
(**A**) Diagrammatic representation of the molecular structure of human large MAF transcription factors, c-MAF, MAFA, MAFB, and NRL. The structural domains are the transactivation domain (in yellow), extended homology region (in green) and basic region (in blue) in the DNA-binding domain, and leucine zipper (in red) and are shown in boxes [5]. (**B**) Consensus sequences recognized by bZip transcription factors. There are two types of responsive elements such as TRE and CRE which are the core sequence of the MARE (in green). They are included within T-MARE and C-MARE, respectively. Three bases on the side of TRE and CRE are the flanking sequence (in red), which plays a critical role in the recognition of the MARE [6].

**Figure 2 genes-14-01883-f002:**
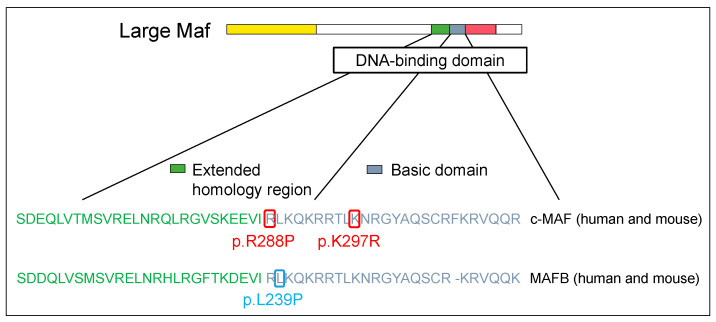
Large Maf DNA-binding domain mutations in human disease. The structural domains are the transactivation domain (in yellow), extended homology region (in green) and basic region (in blue) in the DNA-binding domain, and leucine zipper (in red).

**Figure 3 genes-14-01883-f003:**
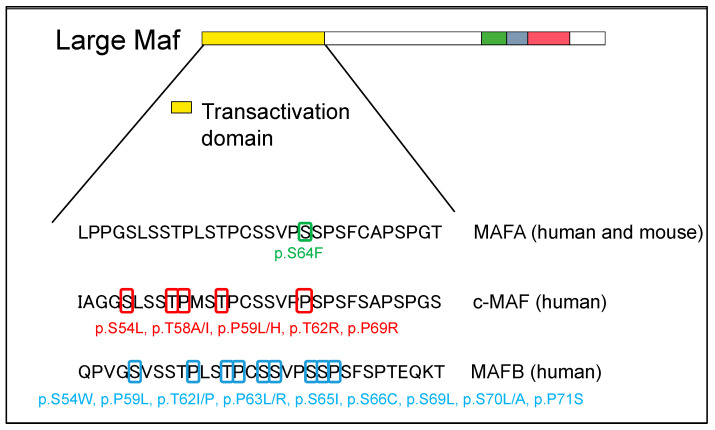
Large MAF phosphorylation site mutations on transactivation domain in human disease. The structural domains are the transactivation domain (in yellow), extended homology region (in green) and basic region (in blue) in the DNA-binding domain, and leucine zipper (in red).

## Data Availability

Not applicable.

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
