# Peer review of "Exploring Large MAF Transcription Factors: Functions, Pathology, and Mouse Models with Point Mutations"

_genes, 2023, doi:10.3390/genes14101883_

Round 1
Reviewer 1 Report
Comprehensive review of human and mouse studies of the roles of transcription factors Large MAFs. Authors made major contribution to the field by creating mouse models in attempts to rule out the link between point mutations in particular memebrs of the MAF family and phenotypic changes in humans and mice. This is a good example of hardcore science that gives definitive answers to biological questions. I appreciate the effort and willingness of authors to share the up to date information concerning large MAF studies. At the same time the manuscript of this excellent review should be improved by making several necessary changes. Below are the points that should be addressed:
1) Scientific English requires extensive editing. Some sentences does not have much sence. For instatnce, legends to figures 1 and 2 have the following sentenses: "The representation is modified a previous figure".
Lines 110, 123, 139 - errant sentences. I could list more, so I insist on editing the text.
2) Figure 4 should be modified. First, human and mouse sequences of MafA/MAFA are identical. You could leave only one, not two. Second, color arrows and boxes should be better organized. It looks really confusing. I would better remove long arrows and replaced them with short vertical arrows and single letter abbreviations of amino acid residue changes due to mutations.
3) Figures 1 and 2 could be united.
4) Lengthy descriptions of point mutations in humans, mice, complex phenotypes are really confusing. I would supplement the text with a table showing mutations in MAF members and resulting phenotypes with links to the literature. It will make the manuscript much easier for readers.
Finally, the verdict. The review could be accepted after the suggested changes will be made.
I suggest changes to figures and organization of the manuscript. Table has to be added, Figure 4 requires corrections. English should be heavily edited, several phrases do not have much sense. Could be accepted after major changes and proofreading (see comments to the authors).
Author Response
Comprehensive review of human and mouse studies of the roles of transcription factors Large MAFs. Authors made major contribution to the field by creating mouse models in attempts to rule out the link between point mutations in particular memebrs of the MAF family and phenotypic changes in humans and mice. This is a good example of hardcore science that gives definitive answers to biological questions. I appreciate the effort and willingness of authors to share the up to date information concerning large MAF studies. At the same time the manuscript of this excellent review should be improved by making several necessary changes. Below are the points that should be addressed:
1) Scientific English requires extensive editing. Some sentences does not have much sence. For instatnce, legends to figures 1 and 2 have the following sentenses: "The representation is modified a previous figure".
Lines 110, 123, 139 - errant sentences. I could list more, so I insist on editing the text.
(Response)
Thank you for your advice. When we submit an academic paper, we ask Editage to proofread our manuscript. To revise our manuscript according to your comments, we saught the services of Editage for English editing again and the revised manuscript has been edited.
2) Figure 4 should be modified. First, human and mouse sequences of MafA/MAFA are identical. You could leave only one, not two. Second, color arrows and boxes should be better organized. It looks really confusing. I would better remove long arrows and replaced them with short vertical arrows and single letter abbreviations of amino acid residue changes due to mutations.
(Response)
Thank you for your suggestion. Accordingly, we modified Figure 4.
3) Figures 1 and 2 could be united.
(Response)
Thank you for your suggestion. We combined Figures 1 and 2.
4) Lengthy descriptions of point mutations in humans, mice, complex phenotypes are really confusing. I would supplement the text with a table showing mutations in MAF members and resulting phenotypes with links to the literature. It will make the manuscript much easier for readers.
(Response)
Thank you for your suggestion. We remade the tables to aid the understanding of the readers regarding the MAF members and resulting phenotypes.
Reviewer 2 Report
The present review by Mitsunori Fujino, Masami Ojima and Satoru Takahashi “Exploring large MAF transcription factors: Functions, pathology, and mouse models with point mutations” attempts to provide a concise summary on sequence-specific DNA-binding transcription factors Maf/c-Maf, MafA and MafB. The major strength of this review is that Takahashi lab has generated a number of seminal insights into the diverse roles of these proteins using mouse genetic studies. However, the authors miss many opportunities to cover this field in more detail and include key original studies published in the last decade, including relevant reviews (Reviews: PMID: 26122665, 25685288, and 21719305).Sections 3. to 11. should be expanded by 4-6 additional pages to provide mechanistic insights into which genes are directly regulated by these transcription factors, how is their tissue-specific gene expression regulated in beta-cells of pancreas, epidermis, lens, macrophages, T-cells, …as well as to describe functional interactions between Mafs and other transcription factors bound in the vicinity (cis-regulatory grammar) within well characterized promoters and enhancers. Taken together, this review manuscript provides interesting though limited insights into structure, function and genetics of these three interesting genes and is within the scope of the Genes.
Additional comments/suggestions:
1) Abstract: Although NRL belongs to this family, it is not discussed in the review like the other proteins/genes and should not be mention in the Abstract but it should be explained why NRL is not discussed in detail and refer to a few appropriate reviews.
2) Some data can be organized with a Table or two.
Author Response
The present review by Mitsunori Fujino, Masami Ojima and Satoru Takahashi “Exploring large MAF transcription factors: Functions, pathology, and mouse models with point mutations” attempts to provide a concise summary on sequence-specific DNA-binding transcription factors Maf/c-Maf, MafA and MafB. The major strength of this review is that Takahashi lab has generated a number of seminal insights into the diverse roles of these proteins using mouse genetic studies. However, the authors miss many opportunities to cover this field in more detail and include key original studies published in the last decade, including relevant reviews (Reviews: PMID: 26122665, 25685288, and 21719305).Sections 3. to 11. should be expanded by 4-6 additional pages to provide mechanistic insights into which genes are directly regulated by these transcription factors, how is their tissue-specific gene expression regulated in beta-cells of pancreas, epidermis, lens, macrophages, T-cells, …as well as to describe functional interactions between Mafs and other transcription factors bound in the vicinity (cis-regulatory grammar) within well characterized promoters and enhancers. Taken together, this review manuscript provides interesting though limited insights into structure, function and genetics of these three interesting genes and is within the scope of the Genes.
(Response)
Thank you for your comment and suggestion. Based on the reviews suggested, we mainly added the roles of each MAF in the revised manuscript as shown below. The point we would like to emphasize most in this review is that the mouse model carrying point mutation that we generated is similar to the human disease in patients carrying these point mutations. Although compact, we believe that this aspect has been sufficiently described with the latest knowledge available citing the review.
Page 3, lines 97–105
For example, the scaffold protein CARMA1 and IKKβ, two essential regulators of the transcription factor nuclear factor κB (NF-κB), activate c-MAF expression. Through the stimulation of the T-cell receptor, increased c-MAF expression results in the production of cytokines [1]. The loss of c-MAF induces a defect in the number of Th-17 and T-cells, suggesting that c-MAF controls the expansion of both Th-17 and T-cells [2]. Subsequently, transcriptome analysis has shown that c-MAF plays an essential role in the differentiation of Th17 [3]. In the case of macrophage, c-MAF controls the expression of IL-10, which is involved with the differentiation of regulatory T cells [4].
Page 4, lines 111–116
Furthermore, c-Maf is regulated by p53 together with Prox-1, and as a result, the expression of various Crystallingenes is regulated [5]. In the case of cell differentiation, c-MAF plays an essential role in the differentiation of lens fiber cells to lens epithelial cells, resulting in epithelial cells spreading in the anterior and posterior lens [6]. Other MAFs, such as MAFA and MAFB, are not required for lens fiber cell differentiation [7].
Page 6, lines 215–220
MAFA coordinates with MAFB, a transactivator of a cells, acting on the glucagon gene G1 element, and in conjunction with other transcription factors and related genes to induce the generation and differentiation of β cells [8,9]. In subsequent studies, MAFB was identified in both α and β cells during the early developmental stage. Following a reduction in MAFB expression, MAFA is mainly expressed instead of MAFB [10,11].
Page 7–8, lines 282–284
Mafb knockout suppressed F4/80 expression in mature macrophages resulting in renal dysgenesis with abnormal podocyte differentiation and tubular apoptosis.
Additional comments/suggestions:
1) Abstract: Although NRL belongs to this family, it is not discussed in the review like the other proteins/genes and should not be mention in the Abstract but it should be explained why NRL is not discussed in detail and refer to a few appropriate reviews.
(Response)
Thank you for your comment. In 2016, a report on a Chinese family with autosomal dominant retinitis pigmentosa was published, wherein the c.146 C>T mutation in the NRL gene was implicated (12), however, to date, no reports on genetically modified mice having the same point mutation have been documented. Therefore, it was not discussed in the review unlike for the other MAF members. We added the following sentences to the Introduction.
Introduction
Page 2, lines 47–50
“This review includes information about existing mouse models, such as those showing the phenotypic features consequent of point mutations in large MAF transcription factors except for NRL because, to date, no reports of genetically modified mice having the same point mutation in the NRL have been documented.”
2) Some data can be organized with a Table or two.
(Response)
Thank you for your suggestion. We merged Figures 1 and 2.
(Reference)
- Blonska M, Joo D, Nurieva RI, Zhao X, Chiao P, Sun SC, Dong C, Lin X. Activation of the transcription factor c-Maf in T cells is dependent on the CARMA1-IKKβ signaling cascade. Sci Signal. 2013 Dec 17;6(306):ra110.
- Bauquet AT, Jin H, Paterson AM, Mitsdoerffer M, Ho IC, Sharpe AH, Kuchroo VK. The costimulatory molecule ICOS regulates the expression of c-Maf and IL-21 in the development of follicular T helper cells and TH-17 cells. Nat Immunol. 2009 Feb;10(2):167-75.
- Sato K, Miyoshi F, Yokota K, Araki Y, Asanuma Y, Akiyama Y, Yoh K, Takahashi S, Aburatani H, Mimura T. Marked induction of c-Maf protein during Th17 cell differentiation and its implication in memory Th cell development. J Biol Chem. 2011 Apr 29;286(17):14963-71.
- Cao S, Liu J, Song L, Ma X. The protooncogene c-Maf is an essential transcription factor for IL-10 gene expression in macrophages. J Immunol. 2005 Mar 15;174(6):3484-92.
- Liu FY, Tang XC, Deng M, Chen P, Ji W, Zhang X, Gong L, Woodward Z, Liu J, Zhang L, Sun S, Liu JP, Wu K, Wu MX, Liu XL, Yu MB, Liu Y, Li DW. The tumor suppressor p53 regulates c-Maf and Prox-1 to control lens differentiation. Curr Mol Med. 2012 Sep;12(8):917-28.
- Kase S, Yoshida K, Sakai M, Ohgami K, Shiratori K, Kitaichi N, Suzuki Y, Harada T, Ohno S. Immunolocalization of cyclin D1 in the developing lens of c-maf -/- mice. Acta Histochem. 2006;107(6):469-72.
- Ring BZ, Cordes SP, Overbeek PA, Barsh GS. Regulation of mouse lens fiber cell development and differentiation by the Maf gene. Development. 2000 Jan;127(2):307-17.
- Matsuoka TA, Zhao L, Artner I, Jarrett HW, Friedman D, Means A, Stein R. Members of the large Maf transcription family regulate insulin gene transcription in islet beta cells. Mol Cell Biol. 2003 Sep;23(17):6049-62.
- Kataoka K, Shioda S, Ando K, Sakagami K, Handa H, Yasuda K. Differentially expressed Maf family transcription factors, c-Maf and MafA, activate glucagon and insulin gene expression in pancreatic islet alpha- and beta-cells. J Mol Endocrinol. 2004 Feb;32(1):9-20.
- Artner I, Le Lay J, Hang Y, Elghazi L, Schisler JC, Henderson E, Sosa-Pineda B, Stein R. MafB: an activator of the glucagon gene expressed in developing islet alpha- and beta-cells. Diabetes. 2006 Feb;55(2):297-304.
- Hang Y, Stein R. MafA and MafB activity in pancreatic β cells. Trends Endocrinol Metab. 2011 Sep;22(9):364-73. doi: 10.1016/j.tem.2011.05.003.
12. Gao M, Zhang S, Liu C, Qin Y, Archacki S, Jin L, Wang Y, Liu F, Chen J, Liu Y, Wang J, Huang M, Liao S, Tang Z, Guo AY, Jiang F, Liu M. Whole exome sequencing identifies a novel NRL mutation in a Chinese family with autosomal dominant retinitis pigmentosa. Mol Vis. 2016 Mar 18;22:234-42.
Round 2
Reviewer 1 Report
Thanks for point-by-point response and corrections in text and figures. No the manuscript can be accepted for publication in its present form. Congrats!
Reviewer 2 Report
N/A